# An Uncertainty-Guided Manifold Smoothing Method for Non-Ideal Measurement Computed Tomography Reconstruction

## Abstract

Non-ideal measurement computed tomography (NICT) reduces the need for extensive data sampling, accelerating scanning and mitigating radiation exposure risks, but these benefits are accompanied by artifacts and noise. While enormous deep learning (DL) methods have been developed to improve image quality, they typically require paired data, which is challenging to obtain owing to physiological motion. Unsupervised reconstruction methods offer a potential solution, but they typically assume homogeneous noise distributions and overlook variations from different sampling strategies, which may cause model collapse. We observe that NICT images form discrete sub-manifolds in feature space due to varying physical scanning processes, which contradicts the assumption of unsupervised methods and limits their effectiveness. To address this, we propose an Uncertainty-Guided Manifold Smoothing (UMS) framework to bridge the gaps between sub-manifolds. In UMS, a classifier is first trained to identify the sub-manifold associated with each feature representation. The predicted uncertainty scores are then used to guide the generation of diverse samples across the entire manifold. By leveraging the classifier's capability, UMS effectively fills the gaps between discrete sub-manifolds, and promotes a more continuous and dense feature space. Given the complexity of the global manifold, it is hard to directly model the manifold. Therefore, we propose to dynamically incorporate the global- and sub-manifold-specific features. Specifically, we design a global- and sub-manifold-driven architecture guided by the classifier, which enables dynamic adaptation to sub-manifold variations. This dynamic mechanism improves the network's capacity to capture both shared and domain-specific features, thereby improving reconstruction performance. Extensive experiments on the public datasets are conducted to validate the effectiveness and generalizability of our method.

## 1 Introduction

Computed tomography (CT) is a non-invasive imaging technique that has been widely used in clinical practice to provide detailed information about internal anatomical structures. However, CT scanning carries the risk of radiation exposure. To alleviate this concern, various sampling strategies have been developed to reduce radiation dose under hardware and scanning constraints, including low-dose CT (LDCT), sparse-view CT (SVCT), and limited-angle CT (LACT). These approaches reduce radiation exposure while also accelerating the scanning process. However, the measured data acquired using these methods are considered non-ideal measurement CT (NICT), and the reconstructed images are often degraded by artifacts and noise, which significantly compromise their clinical applicability.

To improve image quality and ensure clinical utility, numerous methods have been proposed. Among them, deep learning (DL)-based approaches have shown remarkable performance (Chen et al., 2017; Wang et al., 2024). However, most of these methods rely on the availability of paired datasets consisting of NICT and ideal measurement CT (ICT) images for training (Yang et al., 2018; Xia et al., 2023). This assumption is difficult to meet due to the dynamic nature of physiological processes, such as respiration and cardiac motion, which cause organ displacement and make it nearly impossible to acquire paired data.

To address this limitation, unsupervised denoising methods have been developed to eliminate the need for paired data. However, these methods are typically restricted to a single task, such as LDCT (Kwon & Ye, 2021; Kim et al., 2024b), leading to limited generalization and increased deployment overhead. To achieve unified denoising under an unsupervised paradigm, low-quality images from different tasks as a source domain and applying domain mapping. GAN- and diffusion-based approaches are the two mainstream domain mapping methods. CycleGAN (Zhu et al., 2017) is a classic GAN-based method that accomplishes domain mapping through cycle-consistency constraints. GcGAN (Fu et al., 2019) uses a geometry-consistency constraint by inputting an image and its transformed counterpart to reduce the solution space while preserving valid mappings. CUT (Park et al., 2020) improves performance with a patch-based contrastive learning framework that maximizes mutual information between input and output patches. AttentionGAN (Tang et al., 2021) attempts to enhance the generation quality by incorporating an attention mechanism. More recently, UNSB (Kim et al., 2024a) addresses the limitations of diffusion models in unpaired domain mapping by reformulating the Schrödinger Bridge problem as a sequence of adversarial learning tasks. However, most existing methods assume that the data in source domain follows a homogeneous distribution, an assumption that does not hold in the context of NICT reconstruction.

Different non-ideal sampling strategies correspond to different physical imaging processes, leading to variations in imaging pipelines and their associated noise distributions (Yang et al., 2025). For instance, LDCT reduces radiation dose by lowering X-ray intensity, while preserving sufficient angular sampling to maintain diagnostic image quality. In contrast, both sparse and limited-angle sampling achieve dose reduction by decreasing the number of projections, but their acquisition patterns differ: sparse sampling preserves full angular coverage with reduced projection density, whereas limited-angle sampling confines acquisition to a limited angular range. These differences lead to different noise characteristics. To facilitate a clearer understanding, we illustrate the scanning procedures and their corresponding images in Fig. 1, which reveal substantially different noise distributions.

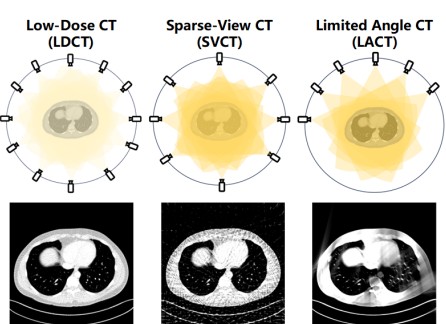

Figure 1: Visualization of reconstruction results under different sampling strategies.

Motivated by these observations, we model the feature representations of images acquired via different non-ideal sampling strategies as points distributed on a complex manifold. Due to fundamentally different physical scanning processes, these points naturally cluster into discrete sub-manifolds corresponding to each scanning strategy. Formally, the manifold of a non-ideal data domain can be modeled as a union of these sub-manifolds. However, these sub-manifolds are often disconnected and discrete, which leads to discontinuities across the global feature space. Such discontinuities pose challenges for learning models, as features from different sub-manifolds exhibit abrupt transitions without smooth and continuous trajectories connecting them.

To address this issue, we propose an Uncertainty-Guided Manifold Smoothing (UMS) framework based on diffusion model for NICT reconstruction. Specifically, a classifier is first trained to identify the sub-manifold associated with each feature representation, and its predictions are used to guide the diffusion process for generating diverse samples across the entire manifold. By incorporating the classifier's uncertainty estimates, particularly near sub-manifold boundaries, the framework effectively bridges the gaps between discrete sub-manifolds. This approach promotes a more continuous and dense feature space, which facilitates smoother transitions across sub-manifold boundaries.

Although the completed data form a global manifold, relying solely on a global representation risks overlooking important sub-manifold-specific characteristics, which compromise the accuracy and robustness of reconstruction. To address this challenge, we design a global- and sub-manifold-driven architecture. This architecture is guided by the same classifier employed during the manifold smoothing stage, which controls the generation process using the confidence scores derived from the classifier. Instead, it leverages the classifier's predictions in the image domain to infer latent relationships among sub-manifolds. Such guidance enables the model to dynamically adapt to sub-manifold-specific features while maintaining consistency with the global manifold, thus benefiting from shared global- and sub-manifold knowledge. The main contributions of this paper can be summarized as follows:

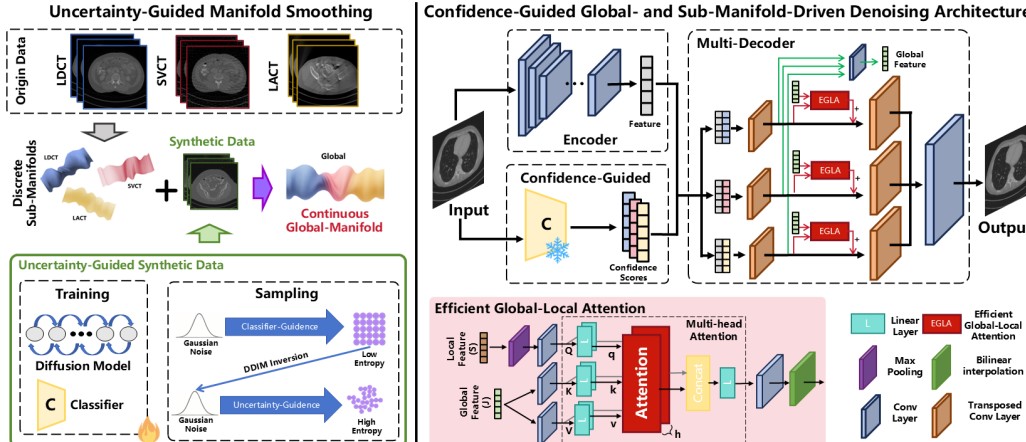

Figure 2: The framework of the proposed method.

- We revisit the unsupervised learning paradigm for NICT reconstruction and propose a novel uncertainty-guided manifold smoothing framework that bridges the gaps between discrete sub-manifolds in the NICT feature space guided by the uncertainty.

- We design a confidence-guided global- and sub-manifold-driven architecture that jointly models and balances the latent relationships between global- and sub-manifold feature representations.

- Extensive experiments on public datasets validate the effectiveness of the proposed method, and the results demonstrate its compatibility with various approaches to further enhance reconstruction performance.

## 2 METHODOLOGY

### 2.1 OVERVIEW

An overview of our method is presented in Fig. 2. The proposed method consists of two stages. The first stage trains a classifier using data from different sub-manifolds. The uncertainty scores produced by this classifier guide the generation of new samples to bridge the gaps between sub-manifolds. Although the completed data form a global manifold, modeling it globally tends to overlook local features. To address this, we propose a confidence-guided global- and sub-manifold architecture in the second stage. This structure leverages the classifier's predictions to infer implicit relationships between sub-manifolds, which enable the network to dynamically adapt to subdomain-specific features. Additionally, a global-local attention module is designed to integrate global- and sub-manifold information effectively.

### 2.2 UNCERTAINTY-GUIDED MANIFOLD SMOOTHING

As mentioned earlier, the proposed smoothing step aims to mitigate the domain gap and enhance the smoothness of sub-manifolds. This module operates in two stages: training and sampling. We now elaborate on its implementation in each stage.

#### 2.2.1 TRAINING STAGE.

The diffusion model is trained by adding Gaussian noise $\epsilon$ to the data $x$ in a forward process, followed by learning to reverse this corruption using a time-conditioned U-Net architecture, which serves as the noise predictor conditioned on label information $y$, with its output denoted as $\epsilon_\theta$. Given a variance schedule $\beta_t$, we can sample the noisy image $x_t$ at any timestep $t$ using the reparameterization trick, where $x_t \sim \mathcal{N}(x_t; \sqrt{\overline{\alpha}_t}x_0, (1-\overline{\alpha}_t)\mathbf{I})$, with $\alpha_t = 1-\beta_t$ and $\overline{\alpha}_t = \prod_{i=1}^{t} \alpha_i$. By incorporating label information during training, the model learns to generate class-related samples (Ho & Salimans, 2022).

The diffusion training loss function is as follows:

$$\mathcal{L}_{\text{diffusion}} = \sum_t \mathbb{E}_{x_0,\epsilon}[W||\epsilon - \epsilon_\theta(x_t, y, t)||^2] + 0.001\mathcal{L}_{vlb}, \tag{1}$$

where $W$ is a time-step-dependent weight that facilitates the learning of visual information, and $L_{vlb}$ introduces a learnable variance into the variational lower bound (VLB) loss to enhance the generative performance of the trained model (Nichol & Dhariwal, 2021).

In addition, implementing guidance sampling requires pre-training a classifier $p_\phi(y|x_t, t)$, which incorporates time information $t$ as input to represent the degree of noise corruption in the noisy image. The classifier is based on the encoder part of a U-Net architecture, which incorporates attention blocks and timestep embeddings. The corresponding training loss function associated with this process is given by:

$$\mathcal{L}_{\text{pre-classifier}} = \sum_t \mathbb{E}_{x_0,\epsilon}[||y - p_\phi(x_t, t)||_c], \tag{2}$$

where $||.||_c$ represents the cross-entropy loss function.

### 2.2.2 CLASSIFIER-GUIDED SAMPLING AND DDIM INVERSION.

To ensure the generated images belong to the specified category, the pretrained classifier guides the generation of high-confidence, class-consistent samples (Dhariwal & Nichol, 2021), after which reverse Denoising Diffusion Implicit Model (DDIM) is applied to derive the corresponding noise data. The noise prediction process is formulated as follows:

$$\hat{\epsilon} = \epsilon_\theta(x_t, y, t) - \sqrt{1 - \bar{\alpha}_t} \cdot \nabla_{x_t} \log p_\phi(x_t, t). \tag{3}$$

Based on the gradient-guided noise prediction, a new sampling process is derived as:

$$x_{t-1} = \sqrt{\bar{\alpha}_{t-1}}(\frac{x_t - \sqrt{1 - \bar{\alpha}_t}\hat{\epsilon}}{\sqrt{\bar{\alpha}_t}}) + \sqrt{1 - \bar{\alpha}_{t-1}}\hat{\epsilon}. \tag{4}$$

To proceed to the next stage of generation, we generate new noisy data by reversing the deterministic generative process of DDIM:

$$x_{t+1} = \sqrt{\bar{\alpha}_{t+1}} \left( \frac{x_t - \sqrt{1 - \bar{\alpha}_t}\epsilon_\theta(x_t, y, t)}{\sqrt{\bar{\alpha}_t}} \right) + \sqrt{1 - \bar{\alpha}_{t+1}}\epsilon_\theta(x_t, y, t). \tag{5}$$

### 2.2.3 UNCERTAINTY-GUIDED SAMPLING.

To improve the diversity of generated samples and promote their distribution near sub-manifold boundaries to ensure a smooth transition across sub-manifolds. Building upon the classifier guidance, we extend it to develop an uncertainty-guided sampling strategy.

From the perspective of stochastic differential equation (SDE), the classifier is more intuitive and naturally supports extending classifier guidance. When the sampling variance is zero, the SDE reduces to ordinary differential equation (ODE), resulting in the following DDIM sampling process

$$dx = \left( f_t(x_t) - \frac{1}{2}g_t^2\nabla_{x_t} \log p_\theta(x_t) \right) dt, \tag{6}$$

where $\nabla_{x_t} \log p_\theta(x_t)$ is score function, $f_t(\cdot)$ is the drift coefficient of $x_t$, $g_t$ is the diffusion coefficient of $x_t$. The conditional information $\mathcal{U}(\cdot)$, derived from the pre-trained classifier $p_\phi(y|x_t, t)$, represents entropy used for uncertainty-guided and is defined as follows:

$$\mathcal{U}(x_t) = -\sum_i p_\phi(y_i|x_t, t) \log p_\phi(y_i|x_t, t). \tag{7}$$

Then, the conditional ODE of uncertainty guidance in DDIM is

$$dx = \left( f_t(x_t) - \frac{1}{2}g_t^2\nabla_{x_t} \log p_\theta(x_t|\mathcal{U}) \right) dt, \tag{8}$$

wherein, exploiting the Bayesian formula in the score function and selecting terms about $x_t$, we obtain:

$$\nabla_{x_t} \log p_\theta(x_t \mid \mathcal{U}) = \nabla_{x_t} \log \left( \frac{p_\theta(x_t)p(\mathcal{U} \mid x_t)}{p(\mathcal{U})} \right)$$

$$= \nabla_{x_t} \log p_\theta(x_t) + \nabla_{x_t} \log p(\mathcal{U} \mid x_t). \tag{9}$$

Next, we leverage the connection between diffusion models and score matching, and the score function can be expressed as:

$$\nabla_{x_t} \log p_\theta(x_t) = -\frac{1}{\sqrt{1 - \bar{\alpha}_t}} \epsilon_\theta(x_t, y, t). \tag{10}$$

Moreover, to obtain sampling results $x_0$ with higher value $\mathcal{U}(x_0)$, we adopt the method from (Luo et al., 2024) and set $p(\mathcal{U}|x_t) \propto e^{\gamma \cdot \mathcal{U}(x_t)}$, where $\gamma$ is a hyperparameter that controls the strength of uncertainty guidance. This leads to $p_\theta(x_t|\mathcal{U}) \propto e^{\gamma \cdot \mathcal{U}(x_t)} p_\theta(x_t)$. Furthermore,

$$\nabla_{x_t} \log p_\theta(x_t \mid \mathcal{U}) = -\frac{1}{\sqrt{1 - \bar{\alpha}_t}} \epsilon_\theta(x_t, y, t) + \nabla_{x_t} \gamma \cdot \mathcal{U}(x_t)$$

$$= -\frac{1}{\sqrt{1 - \bar{\alpha}_t}} \Big( \epsilon_\theta(x_t, y, t) - \sqrt{1 - \bar{\alpha}_t} \cdot \nabla_{x_t} \gamma \cdot \mathcal{U}(x_t) \Big). \tag{11}$$

Analogous to classifier guidance, we introduce uncertainty-guided sampling by replacing $\epsilon_\theta(x_t, y, t)$ in each sampling step with

$$\hat{\epsilon}_\theta(x_t, y, t) = \epsilon_\theta(x_t, y, t) - \sqrt{1 - \bar{\alpha}_t} \cdot \nabla_{x_t} \gamma \cdot \mathcal{U}(x_t). \tag{12}$$

Consequently, the new sampling process tends to generate $x_t$ with higher $\mathcal{U}(x_t)$ values, yielding an $x_0$ with a higher $\mathcal{U}(x_0)$, thereby introducing greater uncertainty to enhance sample diversity.

## 2.3 NETWORK ARCHITECTURE

Although the manifold has been smoothed by the first step, modeling it globally tends to overlook local features. To address this issue, we propose a confidence-guided denoising architecture that leverages both global and sub-manifold information. Our framework consists of two generators and two discriminators. The generator mapping from high- to low-quality domain and the two discriminators follow prior designs. The generator mapping from low- to high-quality domain adopts a novel architecture that models both global- and sub-manifolds. In other words, our method can be seamlessly integrated into existing unsupervised frameworks by simply modifying the generator.

Existing generators for low- to high-quality mapping typically rely on feature processing based on a single, simple manifold, which limits their capacity to capture complex manifolds, such as the smoothed one in our study. To better balance local and global features, we rethink the mapping process from the low-quality to the high-quality domain. These differences of local features primarily arise from noise introduced by different physical scanning procedures. To address this, we model the image in the feature domain as a superposition of multiple sub-manifolds and employ a classifier to obtain confidence scores that implicitly capture their interrelationships. Furthermore, we design multiple decoders, each tailored to the feature processing of different sub-manifolds, and use the confidence scores to guide this process, enabling adaptive feature representation and processing.

Specifically, the generator consists of an encoder and multiple decoders. The encoder extracts features from the input image. To reduce training cost, we reuse the classifier trained within the UMS framework. The classifier assigns a soft label to each image, and the resulting confidence scores are concatenated with the image features as input to the decoder module. The input to each decoder, denoted as $I_d$, is defined as:

$$I_d = Concat(f, c_d), \tag{13}$$

where $f$ denotes the image feature extracted by the encoder, and $c_d$ represents the confidence score for a sub-manifold, reshaped to match the dimensions of $f$. The multiple decoders produce outputs $o_1, o_2, o_3$ corresponding to different sub-manifolds. Finally, the outputs are fused to generate the final high-quality image $O$ as follows:

$$O = Conv(Avg(o_1, o_2, o_3)). \tag{14}$$

To enable more effective feature aggregation across multiple decoder outputs, we introduce a global-local attention module, which allows the sub-manifold-driven decoders to incorporate global feature information during processing. This helps mitigate excessive bias and promotes more reliable aggregation in subsequent stages.

The global-local attention module comprises two parts: global feature representation and efficient global-local attention (EGLA). The global feature representation processes the feature maps from the first transposed convolution layer of all decoders through a convolutional layer to generate a global-manifold representation $J \in \mathbb{R}^{B \times C \times H \times W}$. Each decoder integrates an EGLA module between two transposed convolution layers. The EGLA module receives the global feature $J$ and the intermediate feature map $S \in \mathbb{R}^{B \times C \times 2H \times 2W}$, obtained after the first transposed convolution layer.

The structure of the EGLA module is illustrated in Fig. 2. The core attention mechanism within EGLA takes three inputs: query $q \in \mathbb{R}^{B \times N \times d_k}$, key $k \in \mathbb{R}^{B \times N \times d_k}$, and value $v \in \mathbb{R}^{B \times N \times d_v}$. Here, $B$ represents the batch size, and $N = H \times W$ is the total number of spatial locations. The attention is calculated as follows:

$$\text{Attention}(q, k, v) = \text{softmax}(\frac{qk^T}{\sqrt{d_k}})v. \qquad (15)$$

However, a single attention mechanism captures relationships within one pattern, limiting its ability to model diverse associations. Multi-head attention addresses this by employing multiple parallel heads to capture interactions across different subspaces, thereby enhancing representation capability. We thus adopt the multi-head attention mechanism for feature fusion. The multi-head attention module receives three inputs: query $Q \in \mathbb{R}^{B \times N \times E}$, key $K \in \mathbb{R}^{B \times N \times E}$, and value $V \in \mathbb{R}^{B \times N \times E}$, where $E$ denotes the embedding dimension.

To reduce computational cost, the feature map $S$ from the transposed convolution is downsampled via a max pooling layer, followed by a convolutional layer and flattening operation to generate the query $Q$. Meanwhile, the global feature $J$ is processed by two convolutions and then flattened to generate the key $K$ and value $V$, respectively. Additionally, the attention output is upsampled via interpolation to match the first transposed convolution feature map, enabling a residual connection.

Our confidence-guided global- and sub- manifold-driven denoising network introduces architectural innovations that can flexibly integrate with various loss functions. In this study, we use a conventional cycle-consistency-based loss function to facilitate network training, formulated below:

$$\mathcal{L}(G, D) = \mathcal{L}_{\text{GAN}}(G, D) + \lambda_1 \mathcal{L}_{cyc}(G) + \lambda_2 \mathcal{L}_{identity}(G), \qquad (16)$$

where $G$ and $D$ denote the generator and discriminator, respectively. $L_{GAN}$ represents the adversarial loss used to train the generator and discriminator in an adversarial manner. $L_{cyc}$ denotes the cycle consistency loss, which enforces cross-domain reconstruction consistency. $L_{identity}$ corresponds to the identity loss, which preserves within-domain reconstruction consistency.

## 3 EXPERIMENTS

### 3.1 EXPERIMENTAL SETUP

**Datasets.** The publicly available "2016 NIH-AAPM-Mayo Clinic Low-Dose CT Grand Challenge" dataset (McCollough, 2016) is used to evaluate the effectiveness of the proposed method. To reduce sample redundancy and improve training efficiency, a total of 193 ICT images were selected for training and 46 for testing, where the 193 images were uniformly sampled from the data of eight patients and the 46 images were uniformly sampled from another two patients. Ultimately, our training dataset consists of 579 simulated and UMS-synthesized NICT images as the source domain, and 193 ICT images as the target domain.

**Data Simulation.** Low-quality CT images are generated by undersampling the projection data or introducing noise in the projection domain during simulation. Different scanning strategies were employed to generate downsampled sinogram data, and then FBP was used to reconstruct LDCT, SVCT, and LACT images.

Table 1: Scanning parameters used for simulated CT image generation

| Scanning | Photon Count $\alpha$ | Geometrical Parameters $\beta$ | |
| --- | --- | --- | --- |
| | | View Number | Angle Range |
| LDCT | $1.25e^4$ | 512 | $[0°, 360°]$ |
| SVCT | $1.25e^8$ | 60 | $[0°, 360°]$ |
| LACT | $1.25e^8$ | 512 | $[0°, 125°]$ |

The detailed parameters for each scanning strategy are summarized in Table 1. The reason for not smoothing data through simulation is that each scanning parameter represents a distinct continuous function, exhaustive enumeration is nearly impossible, and the relationships between parameter combinations and the domains they define are difficult to model. Thus, we employ a classifier for guidance, using confidence scores to guide the model toward higher uncertainty.

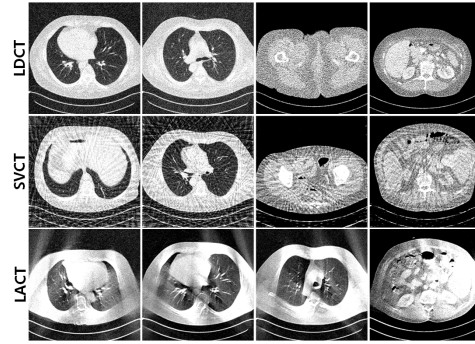

**Data Generation.** To smooth the manifold, the generator and the classifier are trained separately using simulated low-quality data. Then, classifier guidance with a guidance scale of 10 is applied, followed by DDIM inversion to obtain the corresponding noise representation. Subsequently, uncertainty guidance is applied to the noise representation, with a guidance scale of 3. For each sub-manifold, 100 images are generated, and representative samples are presented in Fig. 3.

Figure 3: Generation examples of the uncertainty-guided diffusion model.

**Baselines.** The proposed method is compared against several baseline methods, including CycleGAN(Zhu et al., 2017), GcGAN(Fu et al., 2019), CUT(Park et al., 2020), AttentionGAN(Tang et al., 2021), Switchable-CycleGAN(Yang et al., 2021), SRC(Jung et al., 2022), and UNSB(Kim et al., 2024a).

**Implementation Details.** Following the settings in CycleGAN (Zhu et al., 2017), we set $\lambda_1 = 10$ and $\lambda_2 = 5$, and employed the Adam optimizer with $\beta_1 = 0.5$ and $\beta_2 = 0.999$ to train the proposed network. For the comparison methods, we used the released source codes with their original parameter settings for training and testing. The number of training epochs was set to 200.

**Evaluation Metrics.** Consistent with previous works, Peak Signal-to-Noise Ratio (PSNR) and Structural Similarity (SSIM) are selected to evaluate the performance of different methods. For both metrics, higher values indicate better perceptual and structural image quality.

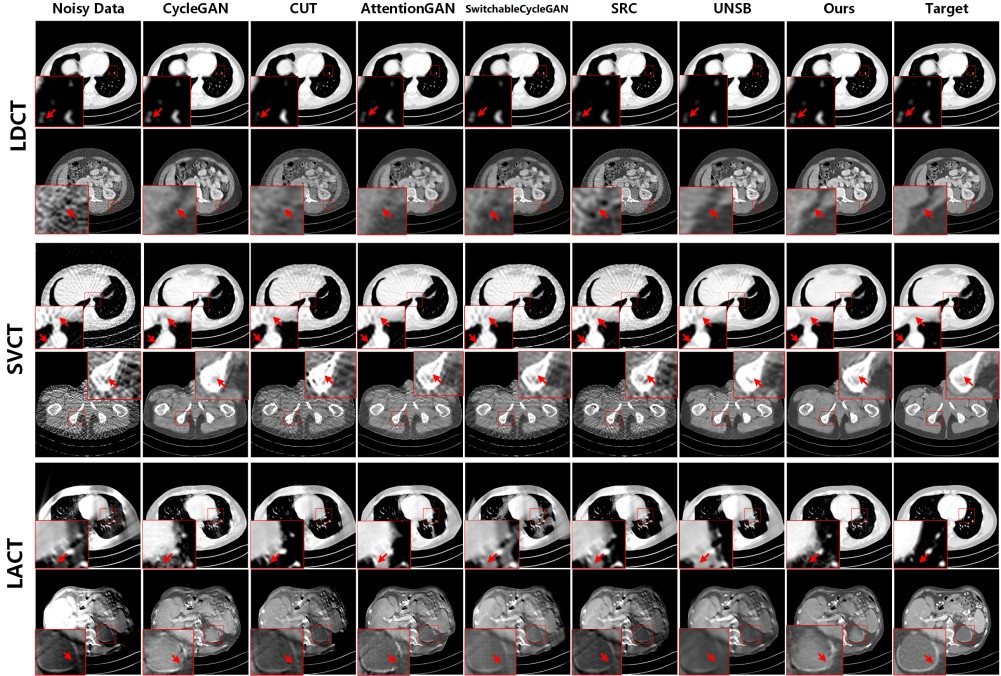

Figure 4: Reconstruction results of different algorithms under varying sampling strategies. The display window for rows 1, 3, and 5 is [-805, 145] HU, while for other rows, it is [-265, 285] HU.

## 3.2 COMPARISON WITH OTHER METHODS

Fig. 4 illustrates the qualitative comparisons across different methods and sampling conditions. While other methods achieve satisfactory performance under specific non-ideal sampling strategies, their generalization to other settings is limited, primarily due to the difficulty of modeling discrete

sub-manifolds within a single unified model. In contrast, the proposed method maintains consistently high performance across all non-ideal sampling strategies.

Quantitative results in Table 2 further support the superiority of the proposed method. Across low-dose, sparse-view, and limited-angle scenarios, the proposed method consistently achieves the best reconstruction performance. The most significant improvement is observed in the

Table 2: Quantitative results of PSNR(dB) and SSIM(%) for different algorithms under varying sampling strategies

|  | LDCT | | SVCT | | LACT | | Average | |
|---|---|---|---|---|---|---|---|---|
| Algorithm | PSNR | SSIM | PSNR | SSIM | PSNR | SSIM | PSNR | SSIM |
| CycleGAN | 38.25 | 94.57 | 36.01 | 91.44 | 31.97 | 90.93 | 35.41 | 92.31 |
| CUT | 36.21 | 90.84 | 33.37 | 81.92 | 29.42 | 83.48 | 33.00 | 85.41 |
| AttentionGAN | 39.59 | 95.83 | 36.70 | 90.91 | 31.01 | 91.06 | 35.76 | 92.60 |
| SwitchableCycleGAN | 37.26 | 92.92 | 33.38 | 84.39 | 27.40 | 85.45 | 32.68 | 87.58 |
| SRC | 35.52 | 88.16 | 32.07 | 79.05 | 29.09 | 83.92 | 32.22 | 83.71 |
| UNSB | 36.82 | 94.81 | 34.81 | 90.93 | 29.29 | 86.77 | 33.64 | 90.83 |
| Ours | 39.66 | 95.18 | 36.89 | 92.34 | 33.79 | 92.06 | 36.78 | 93.19 |

limited-angle scenario, where the proposed method yields PSNR gains exceeding 6 dB compared to competing approaches. Additionally, it achieves notable improvements in average reconstruction quality across all three sub-manifolds.

## 3.3 GENERALIZATION EVALUATION

***Domain Generalization Experiment:*** Due to the uncertainty-guided data generation and the new architecture, the proposed method can adapt to a broader range of data distributions, thereby enhancing its generalization

Table 3: Reconstruction results of different algorithms for domain generalization

|  | Known Domain | | | | Unkown Domain | | Known Domain | | | | Unkown Domain | |
|---|---|---|---|---|---|---|---|---|---|---|---|---|
|  | SVCT | | LACT | | LDCT | | LDCT | | LACT | | SVCT | |
| Algorithm | PSNR | SSIM | PSNR | SSIM | PSNR | SSIM | PSNR | SSIM | PSNR | SSIM | PSNR | SSIM |
| CycleGAN | 35.11 | 89.10 | 31.57 | 90.11 | 38.65 | 94.44 | 36.76 | 90.93 | 31.04 | 89.63 | 31.71 | 78.15 |
| CUT | 32.25 | 77.11 | 28.69 | 81.03 | 36.48 | 91.75 | 35.92 | 87.26 | 28.21 | 81.61 | 31.18 | 73.33 |
| SRC | 30.95 | 65.84 | 28.60 | 82.20 | 34.93 | 82.65 | 34.93 | 82.42 | 28.59 | 81.91 | 30.36 | 70.77 |
| UNSB | 33.11 | 83.33 | 28.51 | 83.38 | 36.57 | 94.10 | 35.64 | 91.87 | 27.69 | 82.40 | 32.03 | 84.31 |
| Ours | 35.88 | 91.55 | 33.46 | 91.32 | 39.00 | 94.79 | 39.53 | 95.39 | 33.15 | 91.35 | 33.46 | 86.06 |

ability. To evaluate the generalization performance on unknown domains, the model is trained on known domains and evaluated on both seen and unseen data distributions. The corresponding NICT image reconstruction results are summarized in Table 3. As shown in Table 3, the proposed method consistently outperforms competing approaches on both known and unseen domains, which demonstrates strong generalization capability even when tested on previously unseen data.

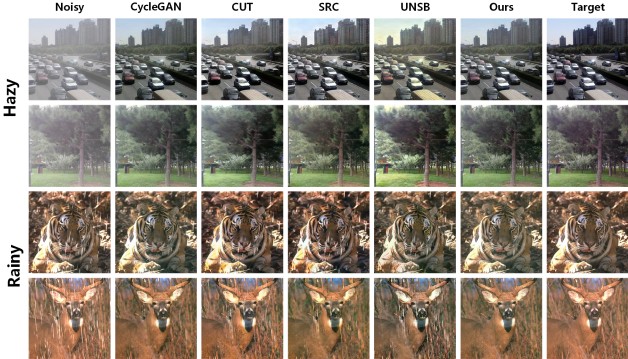

Figure 5: Reconstruction results of different algorithms on natural image dataset.

***Natural Image Experiment:*** Moreover, to demonstrate the broader applicability of the proposed method beyond medical imaging, we evaluate its image reconstruction capability on natural image datasets. Specifically, we consider two natural image reconstruction tasks, image dehazing and image deraining tasks. We construct the low-quality domain using 1,800 lightly rainy images from the JRDR dataset on Kaggle and 500 outdoor hazy images from the SOTS dataset (Li et al., 2019), with their corresponding clean counterparts forming the high-quality domain. To balance the data distribution and validate our method, 1,000 additional hazy images are synthesized using UMS and added to the low-quality domain.

Qualitative comparisons are shown in Fig. 5. In the dehazing task, existing methods often produce images with excessive brightness after haze removal. In contrast, our method not only removes haze effectively but also preserves natural brightness levels, which yields results that are closer to the ground truth. In the deraining task, the proposed method successfully removes most of the streak-like rain while preserving more image details. Besides, the quan-

Table 4: Quantitative results for different algorithms on natural image dataset

|  | Hazy | | Rainy | | Average | |
|---|---|---|---|---|---|---|
| Algorithm | PSNR | SSIM | PSNR | SSIM | PSNR | SSIM |
| CycleGAN | 29.18 | 95.31 | 25.31 | 88.97 | 27.24 | 92.14 |
| CUT | 24.02 | 87.44 | 22.99 | 79.09 | 23.50 | 83.26 |
| SRC | 22.80 | 88.07 | 20.32 | 73.55 | 21.56 | 80.81 |
| UNSB | 22.70 | 83.33 | 21.31 | 72.65 | 22.00 | 77.99 |
| Ours | 29.28 | 95.85 | 25.84 | 90.88 | 27.56 | 93.36 |

titative results are summarized in Table 4. As shown in Table 4, the proposed method consistently outperforms existing approaches in both tasks.

## 3.4 ABLATION EXPERIMENT

***The Effectiveness of UMS:*** The UMS method is designed to construct a smooth manifold that effectively bridges discrete transitions between different sub-manifolds. To evaluate its effectiveness, the data synthesized through the UMS framework were incorporated into the training dataset, and the experimental results of different methods are summarized in Table 5. The results demonstrate that UMS consistently improves the performance of multiple unsupervised image reconstruction methods across diverse domains, with only minor exceptions.

Table 5: Effectiveness of the proposed uncertainty-guided manifold smoothing method

| Algorithm | LDCT PSNR | LDCT SSIM | SVCT PSNR | SVCT SSIM | LACT PSNR | LACT SSIM | Average PSNR | Average SSIM |
|---|---|---|---|---|---|---|---|---|
| CycleGAN | 38.25 | 94.57 | 36.01 | 91.44 | 31.97 | 90.93 | 35.41 | 92.31 |
| CycleGAN$^\dagger$ | 39.25 | 95.30 | 36.32 | 91.95 | 32.84 | 91.76 | 36.13 | 93.00 |
| $\Delta$ | +1.00 | +0.73 | +0.31 | +0.51 | +0.87 | +0.83 | +0.72 | +0.69 |
| Switchable | 37.26 | 92.92 | 33.38 | 84.39 | 27.40 | 85.45 | 32.68 | 87.58 |
| Switchable$^\dagger$ | 38.60 | 94.16 | 34.84 | 88.64 | 27.99 | 86.41 | 33.81 | 89.73 |
| $\Delta$ | +1.34 | +1.24 | +1.46 | +4.25 | +0.59 | +0.96 | +1.13 | +2.15 |
| GcGAN | 40.33 | 96.90 | 36.14 | 92.24 | 29.36 | 89.94 | 35.27 | 93.02 |
| GcGAN$^\dagger$ | 40.13 | 96.73 | 36.64 | 92.43 | 30.34 | 90.39 | 35.70 | 93.18 |
| $\Delta$ | -0.20 | -0.17 | +0.50 | +0.19 | +0.98 | +0.45 | +0.43 | +0.16 |
| SRC | 35.52 | 88.16 | 32.07 | 79.05 | 29.09 | 83.92 | 32.22 | 83.71 |
| SRC$^\dagger$ | 35.13 | 90.34 | 33.29 | 85.77 | 30.32 | 89.78 | 32.91 | 88.63 |
| $\Delta$ | -0.39 | +2.18 | +1.22 | +6.72 | +1.23 | +5.86 | +0.69 | +4.92 |

Notably, UMS provides a "free lunch" improvement, which requires neither architectural modifications nor additional training for the base denoising models. This plug-and-play characteristic makes it a highly versatile and efficient enhancement to existing unsupervised frameworks without introducing computational overhead or implementation complexity.

***Multi-domain v.s. Single-domain Training:*** To assess the comparative advantages of multi-domain versus single-domain training paradigms, a comprehensive experimental study is conducted. The corresponding results are reported in Table 6. The results reveal that joint training consistently improves quantitative performance across multiple domains, with particularly notable gains in the sparse-view and limited-angle scenarios, referred to as CycleGAN (All). CycleGAN$^\star$ refers to a variant in which the generator is replaced with the

Table 6: Ablation study on joint training and contribution parts

| Algorithm | LDCT PSNR | LDCT SSIM | SVCT PSNR | SVCT SSIM | LACT PSNR | LACT SSIM |
|---|---|---|---|---|---|---|
| CycleGAN(LD) | 38.84 | 94.95 | – | – | – | – |
| CycleGAN(SV) | – | – | 35.35 | 90.55 | – | – |
| CycleGAN(LA) | – | – | – | – | 31.91 | 87.92 |
| CycleGAN(All) | 38.25 | 94.57 | 36.01 | 91.44 | 31.97 | 90.93 |
| CycleGAN$^\star$ | 38.51 | 94.19 | 36.22 | 92.11 | 32.51 | 90.36 |
| CycleGAN$^\dagger$ | 39.25 | 95.30 | 36.32 | 91.95 | 32.84 | 91.76 |
| Ours | 39.66 | 95.18 | 36.89 | 92.34 | 33.79 | 92.06 |

proposed confidence-guided global- and sub-manifold-driven architecture. Meanwhile, CycleGAN$^\dagger$ denotes the application of UMS to smooth the manifold.

Besides, the results demonstrate that both components contribute significantly to performance gains over the baseline. Notably, CycleGAN$^\dagger$ yields more pronounced improvements, which aligns with the earlier hypothesis. This observation further supports the claim that the primary challenge lies in modeling discrete manifolds. Once the manifold is smoothed via our proposed approach, reconstruction performance can be substantially enhanced.

## 4 CONCLUSION

In this paper, we propose a novel learning paradigm for unsupervised NICT reconstruction methods that addresses the fundamental challenge of manifold discontinuity across different NICT acquisition strategies. Firstly, we treat the discrete heterogeneous sub-manifold as components of a complex manifold, and design UMS to smooth the manifold. Besides, we design a novel global- and sub-manifold-driven architecture to restore higher quality images through modeling the global- and sub-manifold information simultaneously. Extensive experiments demonstrate our effectiveness. Besides, the smoothing manifold operation can consistently improve the performance for different unsupervised methods without requiring architectural modifications. In future work, extending this methodology to other medical imaging modalities with distinct physical principles represents an interesting research field.

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
