# OpenReview forum: "An Uncertainty-guided Manifold Smoothing Method for Non-Ideal Measurement Computed Tomography Reconstruction"
_ICLR.cc/2026/Conference — ICLR 2026 Conference Withdrawn Submission_

### Official Review · Reviewer_ar7e · 2025-10-26

**Soundness:** 2
**Presentation:** 2
**Contribution:** 2
**Rating:** 2
**Confidence:** 4

**Summary:**

This paper proposes an Uncertainty-Guided Manifold Smoothing (UMS) framework for unsupervised reconstruction of non-ideal measurement CT (NICT) images such as low-dose, sparse-view, and limited-angle CT. The authors argue that data from different sampling schemes form discrete sub-manifolds and use a classifier-guided diffusion process to generate samples bridging these gaps. A confidence-guided architecture further integrates global and sub-manifold features for reconstruction. Experiments on public CT datasets and natural image tasks show improved performance and generalization over existing unsupervised baselines.

**Strengths:**

- The idea of modeling non-ideal measurement CT data as discrete sub-manifolds and addressing discontinuity is conceptually sound.

- The paper provides extensive quantitative and qualitative experiments across multiple domains (LDCT, SVCT, LACT, and natural images).

**Weaknesses:**

- Conceptual ambiguity of the “uncertainty-guided” process: The paper repeatedly emphasizes an “uncertainty-guided” diffusion framework, yet it never defines what uncertainty means in this context or how it differs from conventional classifier-guided diffusion. From the methodology, the approach seems to train a classifier and use its predictions during sampling, but the link between classifier confidence and uncertainty-driven generation is not clearly established.

- Questionable connection between manifold smoothing and diffusion sampling: The claim that diffusion-based training and sampling can smooth discrete sub-manifolds is conceptually weak and lacks theoretical or empirical support. Even if we accept the notion of distinct sub-manifolds for different CT acquisition types, it remains unclear how diffusion sampling achieves genuine smoothing or why this would improve reconstruction.

- Structural and conceptual disconnection between sections: The proposed “manifold smoothing” stage (Sec. 2.2) and the “global- and sub-manifold-driven” reconstruction network (Sec. 2.3) appear largely decoupled. The paper does not clarify whether the diffusion-generated samples are actually used in network training or merely serve as conceptual motivation. As a result, the workflow feels fragmented, and the overall contribution is hard to trace.

- Unclear technical details: Several important design elements remain unspecified—for example, whether the classifier and diffusion model is trained on high-quality or low-quality CT images, what the label $y$ represents, and how uncertainty is computed and utilized in practice. The mathematical exposition is dense but lacks sufficient intuition to understand the practical mechanism.

- Limited novelty and practical insight: The method’s core (classifier-guided diffusion) closely resembles existing guided diffusion frameworks, with limited clarification of what is genuinely new. Furthermore, the two-stage pipeline involving both diffusion and reconstruction networks likely incurs high computational cost, yet the paper provides no analysis of runtime, memory, or scalability—important factors for CT applications.

**Questions:**

Please see the weakness section.

---

### Official Review · Reviewer_VRh2 · 2025-10-29

**Soundness:** 2
**Presentation:** 1
**Contribution:** 2
**Rating:** 2
**Confidence:** 4

**Summary:**

The paper proposes Uncertainty-guided Manifold Smoothing for reconstructing computer tomography data suffering from various types of data limitations: low-dose CT, sparse-view CT, and limited-angle CT. The method uses a classifier and diffusion model to bridge gaps between the discrete sub-manifolds of the various data limitations. Results are shown for simulated data from a single setting in CT, with some additional results for non-CT problems.

**Strengths:**

- Connecting different data limitations in CT with each other could be a fruitful approach.
- Results are compared with some baselines.

**Weaknesses:**

- Non-ideal measurement computed tomography has a very long history of research associated with it. Unfortunately, the paper does not mention or compare with a large body of works, especially those specifically designed for challenging CT cases. Such methods should be mentioned and referenced in the introduction, and representative methods should be compared with in the experimental section to establish how the proposed method compares. Examples of existing approaches include diffusion methods for inverse problems (e.g. [1]. [2], [3]), INR - based methods (e.g. [4]), and CT-specific self-supervised methods (e.g. [5], [6]). In addition, there are many classical approaches, such as TV-minimization, that are used in practice.

- Results are only given for a single dataset (the Mayo clinic data) with simulated data, which does not mean 'extensive experiments on public datasets' as the authors claim. Results for more datasets would strengthen the paper, especially if non-simulated data would be included.

- The required computation time (and how it compares with other methods) is not clearly presented in the paper, even though this is quite important for real-world applications of CT.

- It is not clear how specific hyperparameters settings (e.g. '0.001' and 'W' in equation 1) are chosen by the authors, and how results are affected by different choices for the hyperparameters. This is also try for comparison methods -- how are the hyperparameters chosen for these?

- I find the overall structure of the paper unclear. For example, it is not clearly motivated why combining the submanifolds could lead to better results than treating each submanifold separately.

[1] Zirvi, R., Tolooshams, B., & Anandkumar, A. (2024). Diffusion state-guided projected gradient for inverse problems. arXiv preprint arXiv:2410.03463.

[2] Song, B., Kwon, S. M., Zhang, Z., Hu, X., Qu, Q., & Shen, L. (2023). Solving inverse problems with latent diffusion models via hard data consistency. arXiv preprint arXiv:2307.08123.

[3] Chung, H., Kim, J., Mccann, M. T., Klasky, M. L., & Ye, J. C. (2022). Diffusion posterior sampling for general noisy inverse problems. arXiv preprint arXiv:2209.14687.

[4] Molaei, A., Aminimehr, A., Tavakoli, A., Kazerouni, A., Azad, B., Azad, R., & Merhof, D. (2023). Implicit neural representation in medical imaging: A comparative survey. In Proceedings of the IEEE/CVF International Conference on Computer Vision (pp. 2381-2391).

[5] Baguer, D. O., Leuschner, J., & Schmidt, M. (2020). Computed tomography reconstruction using deep image prior and learned reconstruction methods. Inverse Problems, 36(9), 094004.

[6] Hendriksen, A. A., Pelt, D. M., & Batenburg, K. J. (2020). Noise2inverse: Self-supervised deep convolutional denoising for tomography. IEEE Transactions on Computational Imaging, 6, 1320-1335.

**Questions:**

- How does your method compare with the large body of existing works for non-ideal measurement CT?

- How does your method perform on other datasets, especially non-simulated ones?

- What are the computational requirements of your method?

- How were hyperparameters chosen, and how do hyperparameters affect results?

---

### Official Review · Reviewer_2EGa · 2025-10-30

**Soundness:** 2
**Presentation:** 1
**Contribution:** 1
**Rating:** 2
**Confidence:** 4

**Summary:**

The paper proposes a  uncertainty-guided manifold smoothing framework to bridge the gap between the sub-manifolds associated with different non-ideal CT measurements. This is used for CT reconstruction.

**Strengths:**

- The uncertainty-guidance idea is interesting.

**Weaknesses:**

- The premise of the whole paper is unsound. If the sub-manifolds associated with different NICT schemes ( LDCT, SVCT, LACT considered here) are as disjoint as mentioned here, then wouldn't classifier-free guidance learn to cluster these accurately? In fact, the description (and the images associated with each of the NICTs) suggests that it should be easy to pre-train a classifier that is almost perfectly accurate, not requiring all these smoothing issues.
- Similarly, why isn't this solved as an inverse problem that incorporates the forward operator? This way, one would only need to train an unconditional prior on CT images, and adapt to each different NICT through the sampling process. I understand CT literature focuses on "image enhancement" type solutions, but they do this knowingly with specialization for each type of NICT. To have a broad solver, the inverse problem formulation seems more apt.
- The arguments surrounding manifolds/sub-manifolds are too hand-way. Please provide some (theoretical) justification for the statements.
- Luo et al (2024) has proposed a very similar idea with measurement guidance instead of uncertainty guidance, which makes sense in their context. This is adapted to uncertainty, but that comes with some issues: For instance U as defined in (8) is a function of x_t, yet its log-derivative is set to 0 in (9).
- Similarly, the network is trained on noisy manifolds with no information from uncertainty, so why would this sampling stay on the noisy manifolds seen during training? This is in contrast to measurement, where we do want explicit measurement guidance, potentially on the tangent space. Uncertainty as defined in (8) doesn't have this natural "guidance" perspective.
- I do not understand the point of Section 3.3. Why should the pretrained model work on other sub-manifolds? Isn't this against the hypothesis that these sub-manifolds are discrete/separated?
- Similarly, why would they even work on natural image data for completely different dehazing problem? Were they retrained in this setup?

Minor points:
- The authors get to the relationship between SDE and PF-ODE in a strange manner. Nominally g_t contains information about the noise level, so I am not sure what they mean here.
- Multiple decoders are designed, but this sounds like a multi-task learning problem. What is the benefit of running multiple decoders instead of sharing some layers?
- The method still requires reference target data, so I'm not sure if it should be called "unsupervised" in the inverse problem sense.

**Questions:**

These were already in the weaknesses:
- Why wouldn't a class-conditioned guidance be sufficient to discriminate among these 3 sub-manifolds? Why can't we train a perfect classifier on these 3 classes if they are so well-separated/dscrete?
- Why isn't the NICT problem for multiple different forward operators solved as an inverse problem that incorporates the forward operator?
- U as defined in (8) is a function of x_t, why is its log-derivative is set to 0 in (9)?
- The network is trained on noisy manifolds with no information from uncertainty, so why would this sampling stay on the noisy manifolds seen during training? This is in contrast to measurement, where we do want explicit measurement guidance. What is the theoretical justification for the uncertainty guidance in this case?
- For Section 3.3: Why should the pretrained model work on other sub-manifolds? Isn't this against the hypothesis that these sub-manifolds are discrete/separated?
- Similarly, why would they even work on natural image data for completely different dehazing problem? Were they retrained in this setup?

---

### Official Review · Reviewer_GFcG · 2025-11-01

**Soundness:** 2
**Presentation:** 2
**Contribution:** 2
**Rating:** 4
**Confidence:** 3

**Summary:**

This paper addresses unsupervised CT reconstruction under non-ideal measurement CT (NICT) settings. The key observation is that different acquisition strategies induce discrete sub-manifolds in feature space, making a single unsupervised mapping brittle. The authors propose UMS to train a time-aware classifier over sub-manifolds and use its uncertainty (entropy) to guide diffusion sampling so as to generate bridge samples near sub-manifold boundaries; and invert DDIM to obtain noise and perform uncertainty-guided sampling with a scalar $\gamma$ to bias toward higher-uncertainty regions to densify the manifold. They then design a confidence-guided globa and sub-manifold architecture: a shared encoder, multiple decoders aligned to sub-manifolds, and a global-local attention (EGLA) that fuses a global manifold feature with decoder features; training uses CycleGAN-style losses. Experiments on simulated NIH-AAPM Mayo slices report PSNR/SSIM gains compared to previous methods.

**Strengths:**

1. The motivation of solving non-ideal CT reconstruction is good, and the manuscript is generally well organized.
2. Modeling NICT heterogeneity as sub-manifolds is a crisp lens on why unified unsupervised methods struggle. The uncertainty-guided sampling is an elegant way to populate the gaps.
3. The paper checks many unsupervised baselines, includes domain-generalization setups (train on some sub-manifolds, test on seen & unseen), and ablates UMS as a plug-in to other frameworks.

**Weaknesses:**

1. All NICT data appear simulated from Mayo with fixed geometries and parameter choices; no real LDCT / sparse / limited-angle acquisitions (or realistic patient motion) are used. Claims of generalization across acquisition physics thus remain unproven on real scanners or third-party sites.
2. The manuscript asserts UMS bridges sub-manifold gaps, but provides no feature-space evidence (e.g., t-SNE/UMAP of encoder features, inter-cluster distances, density metrics pre/post-UMS). Current support is performance-based rather than geometric.
3. The text claims “>6 dB PSNR improvement in the limited-angle scenario,” whereas Table 2 shows LACT PSNR 33.79 vs the strongest baseline ~31.97 (≈ **+1.8 dB**, not +6).

**Questions:**

Please refer to the weakness part. I would like to read the rebuttal and increase my rating if my concerns are adequately addressed.

---

### Note · Authors · 2025-11-12

I have read and agree with the venue's withdrawal policy on behalf of myself and my co-authors.